# Skin-to-Skin Care and Spontaneous Touch by Fathers in Full-Term Infants: A Systematic Review

**DOI:** 10.3390/bs14010060

**Published:** 2024-01-17

**Authors:** Laura Cordolcini, Annalisa Castagna, Eleonora Mascheroni, Rosario Montirosso

**Affiliations:** 0–3 Center for the at-Risk Infant, Scientific Institute IRCCS “Eugenio Medea”, 22040 Bosisio Parini, Italy; laura.cordolcini@lanostrafamiglia.it (L.C.); annalisa.castagna@lanostrafamiglia.it (A.C.); eleonora.mascheroni@lanostrafamiglia.it (E.M.)

**Keywords:** fathers, paternal tactile behaviors, skin-to-skin care, spontaneous touch, caregiving

## Abstract

A series of studies have shown that mothers’ early tactile behaviors have positive effects, both on full-term and preterm infants, and on mothers alike. Regarding fathers, research has focused mostly on paternal skin-to-skin care with preterm infants and has overlooked the tactile behavior effects with full-term newborns on infants’ outcomes and on fathers themselves. The current systematic review considered the evidence regarding paternal tactile behaviors with full-term infants, including skin-to-skin care (SSC) and spontaneous touch (ST), during parent–infant interactions, and differentiated biophysiological, behavioral and psychological variables both in fathers and in infants. We also compared fathers’ and mothers’ tactile behaviors for potential differences. The few available studies suggest that paternal touch—SSC and ST—can have positive effects on fathers and infants alike. They also show that, despite some intrinsic differences, paternal touch is as pleasant as maternal touch. However, given the paucity of studies on the topic, we discuss why this field of research should be further explored.

## 1. Introduction

Early tactile parent–infant interactions are very important and play a central role in an infant’s physical, social and emotional development [1]. During early parent–infant interactions, countless exchanges involve tactile contact behaviors, which are expressed through skin-to-skin contact (i.e., a parent holding his/her infant and providing affectionate touch, such as caresses, hugs, and kisses) and are not necessarily related to a particular aim. A plethora of studies have documented that mothers’ early tactile contact behaviors have positive effects both on full-term [2,3] and preterm infants [4,5], while research on paternal tactile behaviors is still scant and its association with infant outcomes remains poorly investigated, especially in full-term infants. This is likely due to the fact that mothers historically spent more time in caregiving compared to fathers [6], although this has now changed, especially in Western societies [7], and nowadays fathers spend much more time caring for, and interacting with, their children than in the past [8]. Findings from industrialized countries reveal that there has been a three- to six-fold increase in father engagement in caregiving activities in the last fifty years [9,10]. Fathers spend much more time not only in caregiving routines (e.g., diaper changing, feeding, playing, …) [11,12], but also in a countless number of embodied interactions [13]. A variety of fathers’ behaviors and their effect on infants’ development have been observed, for example, attachment [14], speech [15], play [16], sensitivity [17], and fathers’ representation of their children [18], and tools have been developed to assess the quality of father–infant interaction [19]. Nevertheless, paternal touch behaviors have remained largely unexplored.

Most studies on fathers’ touch were conducted with preterm infants and their fathers, with a particular focus on skin-to-skin care (SSC). SSC is a practice where a naked infant is placed on his/her parent’s bare chest. It is commonly used immediately after birth with full-terms, or any time an infant needs to be comforted or calmed down. If necessary, infants can wear a diaper and/or a cap and parents and infants can be covered with a blanket or linen. Importantly, given that it is a very low-cost practice that does not require specific materials and equipment, it is suitable for all cultural backgrounds [20]. Research has shown several effects of SSC on preterm infants and their fathers, such as stress reduction [21], changes in paternal oxytocin levels [22] and effectiveness in neonatal pain control [23]. Previous reviews demonstrated that SSC with fathers has a beneficial impact on infants (both pre- and full-terms) and fathers’ outcomes [24,25]. Nevertheless, these studies did not consider any other paternal touch behaviors such as spontaneous touch (i.e., all behaviors used by parents to touch infants for different reasons such as playing, cleaning, getting attention, affection expression), and only a limited number of studies, e.g., [26,27,28], have investigated spontaneous touch (ST). Moreover, to the best of our knowledge, available research on ST touch in full-term infants has not been systematically reviewed yet. Therefore, based on this background, the aims of the current work were two-fold:(1)The primary aim of this review was to systematically analyze paternal touch behaviors with full-term infants, including SSC and ST, during parent–infant interactions. Specifically, the current review focuses on the impact of father–infant SSC on both, including biophysiological markers (e.g., oxytocin and heart rate), behavioral responses (e.g., crying and breastfeeding) and paternal psychological variables (e.g., paternal stress, depression, and bonding).(2)The second aim was to examine the available evidence from comparative studies looking at potential differences between fathers and mothers in early tactile caregiving behaviors.

## 2. Materials and Methods

### 2.1. Search Strategy

A computerized literature search was run for studies published between 2010 and 2023 in the following databases: PubMed and Web of Science. This time period was chosen to ensure the novelty of the review and because fathers have only recently been involved in perinatal care [8]. The following terms were used for all searches: “father AND touch” OR “father-infant AND touch” OR “father AND skin-to-skin” OR “father-infant interaction”. A manual search was also conducted to identify other eligible papers. The Preferred Reporting Items for Systematic Reviews and Meta-Analyses (PRISMA) guidelines [29] were used.

### 2.2. Inclusion and Exclusion Criteria

All papers identified through the search strategy were reviewed (LC and AC) to determine whether studies met the following inclusion criteria: (a) papers written in English, (b) papers published from 2010 onwards in indexed journals in order to guarantee novelty, (c) studies including only fathers or comparing fathers and mothers, and (d) articles on healthy full-term infants/toddlers. Exclusion criteria included: (a) studies not focusing on touch or physical proximity as a mode of interaction, (b) papers on fathers with documented mental disease, (c) reviews, (d) protocol studies and single case studies.

### 2.3. Quality Appraisal

The methodological quality of the papers selected to be included in our review was evaluated using the Quality Assessment Tool for Quantitative Studies [30], in particular, sections A–F (A—selection bias; B—study design; C—confounders; D—blinding; E—data collection methods; F—withdrawals and drop-outs). Two independent authors (LC and EM) coded them as 3 (weak), 2 (moderate), or 1 (strong) according to the component rating scale criteria. A 98% agreement was reached for the A–F components. Disagreements were generally due to different interpretations and they was discussed and resolved under the supervision of a third author (AC). A final 1–3 score was assigned to each paper according to the presence of 2 or more weak scores (3—weak), only 1 weak score (2, moderate), or no weak scores (1—strong). Overall, *n* = 3 studies (20%) were classified as strong, *n* = 9 (60%) as strong to moderate and only 3 studies (20%) as weak. Specific and final ratings are shown in Table 1.

### 2.4. Study Selection and Data Collection Processes

The final pool of studies was obtained as shown in Figure 1 as follows: (1) screening for language other than English; (2) screening for target populations, excluding studies not including fathers or father–mother comparisons; (3) screening for core topics, excluding studies that did not consider paternal touch and/or physical closeness; and (4) inclusion of studies identified by a more extensive manual search [33]. A total of 14 papers meeting the inclusion criteria were identified.

## 3. Results

### 3.1. Data Synthesis, Analysis and General Methodological Approach

All eligible papers were examined by an author (LC) and grouped into 1. studies focusing on SSC; and 2. studies in which fathers touch their infants freely and spontaneously during an interaction. Some studies compared tactile behaviors between mothers and fathers. Then, based on the two main groups (i.e., SSC and ST), papers were further categorized according to the focus of tactile stimulation effects (i.e., biophysiological markers and behavioral variables) either on fathers or on infants. The paper classification is shown in Figure 2.

As mentioned above, papers were divided into three areas: SSC (*n* = 7), ST (*n* = 5) and father–mother comparison (*n* = 7). Samples participating in the selected studies range from 35 [28] to 272 [33]. Children’s ages range from a minimum of 1 month (i.e., newborns) to a maximum of 12 months. All of the studies included both males and females and included participants from different cultural backgrounds. Observed variables include biophysiological markers (7 studies), psychological variables (4 studies), and behavioral responses (7 studies). Four studies have a longitudinal design and 11 studies have a cross-sectional design. A summary of selected papers is reported in Table 2.

### 3.2. Skin-to-Skin Care

#### 3.2.1. Paternal Outcomes

##### Biophysiological Measures

One study reports fathers’ oxytocin, cortisol, and testosterone responses to their first holding of their infants, comparing standard routine holding with SSC holding [41]. Oxytocin was higher after first holding in both groups and there was no difference in oxytocin levels following routine holding vs. SSC holding. Furthermore, fathers whose testosterone increased and oxytocin decreased during first holding showed greater involvement, more direct father–infant caregiving and greater father–infant bonding compared to fathers whose testosterone and oxytocin declined [41].

##### Behavioral Variables

Two studies analyzed vocalization behavior during SCC contact. Velandia and colleagues [32] found that fathers directed more soliciting sounds and speech to their infants during SCC compared with routine care. In another study, Velandia and collaborators [34] documented that fathers appeared to address less speech to girls vs. boys during SSC.

##### Psychological Variables

Most research focused on psychological variables, meaning fathers’ experienced attachment to their infants. In only one study, psychological variables were anxiety and depression [38]. Moreover, most studies compared father–infant dyads during SSC to father–infant dyads during standard care (i.e., simply holding their infants without skin-to-skin contact). Chen and colleagues [35] recruited 92 fathers and their full-term newborns and divided the sample into two groups: fathers during SSC and fathers during standard care. Fathers filled out the Father–Child Attachment Scale [43], a questionnaire measuring attachment on a number of subscales (exploring, touching, caring, and talking) before and after SSC and standard care (i.e., three days after SSC or standard care). Subscale scores post-SCC/standard care were significantly higher in the SSC group than in the standard care group. The difference in total score pre- vs post-SCC/standard care was higher for the SSC group than in the standard care group.

Similarly, one study recruited full-term healthy infants born by caesarean section and fathers were divided into two groups: SSC vs. routine care [38]. Fathers filled out self-report questionnaires about anxiety, depression, and role attainment. Fathers who engaged in SSC had lower anxiety/depression and higher paternal role attainment scores after SSC. Yilmaz and colleagues [42] showed that similar findings are true also in a middle-term perspective. They compared father–infant attachment—as measured by the Paternal–Infant Attachment Scale [44]—in fathers during SSC immediately after birth and fathers who did not engage in SSC 6-to-12 months after birth. Attachment was more intense in fathers who engaged in SSC, especially in first-time fathers.

#### 3.2.2. Infant Outcomes

##### Biophysiological Measures

In Huang et al.’s study [38], some children were exposed to SSC, others received routine care in an incubator next to their father. Infants who received SSC had more stable heart rates and a significantly higher forehead temperature. Similarly, full-term newborns born by elective caesarean section who experienced SSC exhibited higher and more stable heart rates compared with infants who received routine care (i.e., placed in a crib) or babies placed in their fathers’ arms [40].

##### Behavioral Response

Huang et al. [38] also collected data about crying and when children were breastfed and for how long. Infants in the SSC group cried for a shorter time period and showed breastfeeding behaviors earlier than infants in the routine care group. Ayala et al. [40] compared infants placed in a cot and in their fathers’ arms with infants who were exposed to SSC. The latter exhibited higher wakefulness (i.e., the condition of being alert, rather than sleepy) as assessed by the Neonatal Behavioral Assessment Scale (NBAS, [45]).

### 3.3. Spontaneous Touch

Spontaneous touch in the included studies was observed only with reference to fathers’ and infants’ biophysiological markers (e.g., heart rate, respiratory rate, and hormonal levels).

#### 3.3.1. Paternal Outcomes

One study measured paternal oxytocin and coded fathers’ touch during a free play interaction with their 6-month-old infants [26]. They found that physical touch, particularly playful proprioceptive touch, is associated with higher oxytocin levels in fathers. Tactile behaviors, such as cradling, affectionate touch, proprioceptive touch, and stimulatory touch, were coded during interaction between fathers and their 4–6-month-old infants [31]. Only fathers exhibiting high levels of stimulatory contact showed an oxytocin increase.

One study examined the effects of oxytocin administration on paternal behavior and its effects on interactions between fathers and their 5-month-old infants [28]. Participants were divided into two groups: one received intranasally administered oxytocin, while the other received a placebo. Participants were blind to the group they belonged to. Fathers who received oxytocin exhibited more infant-directed touch, positive vocalizations, and encouragement of infants’ social initiative compared to fathers receiving the placebo. Gordon and colleagues [36] explored the interaction between testosterone and oxytocin during transition to parenthood immediately after birth in fathers (and mothers). At 1- and 6-months postpartum, they assessed plasma testosterone and oxytocin concentrations, and they microcoded father–infant interactions of affectionate touch and parent–infant synchrony (i.e., parent engagement in social gaze, affectionate touch, and “baby-talk” vocalizations while the infant looked at the parent and expressed positive affect). Results highlighted that, only when testosterone was high, did negative associations emerge between oxytocin and father affectionate touch.

#### 3.3.2. Infant Outcomes

Cardiac activity (i.e., ECG) and breathing movements before, during, and after a stroking period were collected in 4–16-week-old full-term infants [39]. Findings suggested that paternal gentle stroking induced an increase in respiratory sinus arrhythmia (RSA, an index of cardiac vagal activity) in infants.

### 3.4. Comparison between Fathers and Mothers

#### 3.4.1. Paternal Outcomes

In several studies, Feldman and colleagues [31,33,36] examined the association between endocrine and hormonal biomarkers (oxytocin and testosterone concentrations) and mothers and fathers’ touch behaviors in association with playful interactions with their infants during the postpartum period. Regardless of mother/father and type of touch (e.g., affectionate, functional, stimulatory, and accidental touch), parents with high plasma oxytocin touched their infants more than parents with low oxytocin [33]. Notwithstanding this, significant differences emerged in another study that assessed oxytocin levels in mothers and fathers after they had been engaged in a 15-min play-and-contact interaction with their 4–6-month-old infants [31]. Although baseline oxytocin (plasma and salivary) levels in mothers and fathers were similar, oxytocin was associated with a parent-specific mode of tactile contact. An oxytocin increase after mother–child interaction was observed only in mothers who provided high levels of affectionate contact (e.g., kisses, caresses, and light pokes). Only fathers exhibiting high levels of stimulatory contact (e.g., changing infants’ position in space) showed an oxytocin increase.

Along with oxytocin, testosterone has important implications for the development of social attachment [46]. One study examined potential interactions between these hormones and the development of mothering and fathering in the months postpartum [36]. Specifically, the authors investigated how circulating levels of oxytocin and testosterone were related to affectionate touch and parent–infant synchrony during 5-min play interactions in 1–6-month-old infants. A positive association was found between oxytocin and affectionate touch among mothers with high testosterone levels; in contrast, high testosterone levels in fathers provided the background for negative associations between oxytocin and paternal touch [36].

#### 3.4.2. Infant Outcomes

One study compared the impact of paternal and maternal affectionate touch on infants’ physiological regulation in terms of RSA, which reflects the specific component related to the parasympathetic inhibitory influence on the heart mediated by the vagus nerve [39]. Parental touch behavior was observed during a 3-min affectionate touch period and compared with the baseline and poststroking periods in a group of mothers and fathers and their 4–16week-old infants. Results showed that both mothers’ and fathers’ stroking speed occurred within the optimal stimulation range of C-tactile (CT) afferents, a specific class of cutaneous unmyelinated, low-threshold, mechano-sensitive nerves hypothesized to be involved in bonding [47]. Importantly, no significant difference between the impact of paternal and maternal affective touch on RSA was found. This suggests that parental affective touch has a beneficial impact on parasympathetic infant regulation, regardless of whether it comes from mothers or fathers.

Another study examined the potential differences in negative emotionality between infants who experienced maternal SSC and those who experienced paternal SSC [32]. Although paternal SSC was associated with less crying, infants whined more with fathers than mothers. No significant differences emerged in the quiet state time. A similar procedure analyzed breastfeeding and crying behaviors in newborns, highlighting that fathers touched their infants significantly less than mothers during SSC and breastfeeding started later when SSC was provided by fathers [34]. Another study enrolled full-term infants born by elective caesarean section [37]. Infants were divided into three groups, depending on what happened immediately after surgery: maternal SSC, paternal SSC, and no SSC. Participants were asked to answer an ad hoc interview about infants’ breastfeeding habits immediately after birth and 3 and 6 months later. Immediately after birth, infants who received maternal SSC were more likely to receive exclusive breastfeeding (i.e., the infant receives only breast milk; no other liquids or solids are given—not even water—with the exception of oral rehydration solution, or drops/syrups of vitamins, minerals or medicines) than infants who received paternal SSC or routine care. These results were replicated at three and six months, even if the percentage of infants receiving exclusive breastfeeding dropped.

## 4. Discussion

This review examined studies which used several approaches such as biophysiological, psychological, and behavioral methodologies, to look at the impact of paternal SSC and ST during parent-infant interactions in full-term healthy infants. We also included studies comparing fathers and mothers in their tactile behaviors with their infants. We found that studies considering this research topic are still scant.

Despite this, it emerged that paternal touch could have positive effects on both fathers and infants, both with regard to SSC and ST. Indeed, there is a relationship between paternal biophysiological markers, such as oxytocin, and father–infant tactile interactions [26,33,36,41], although this relationship can change in light of the experimental design. Some evidence suggests that higher basal oxytocin predicts a higher number of paternal tactile behaviors [33], while other evidence shows that tactile behaviors increase the levels of postinteraction oxytocin [26]. This association could be influenced by other significant paternal variables (i.e., parity status and testosterone levels). In particular, in Gettler et al. [32] it was reported that first-time fathers’ oxytocin was higher following first holding of their newborns, compared to their preholding levels. Gordon et al. [43] reported that the association between oxytocin and father–infant tactile interaction (i.e., affectionate touch) was moderated by another biophysiological marker (i.e., testosterone); only in the fathers with high levels of testosterone was oxytocin negatively associated with paternal affectionate touch. Furthermore, fathers involved in SSC engaged more in vocal communication with their full-term infants, especially with their infant boys [34]. Lastly, like mothers involved in SSC [48], and similarly to fathers who provided affectionate ST [26], fathers benefitted in terms of attaining a paternal role and better interaction with their infants, and were less stressed and anxious [32,34,38,49]. Overall, these results demonstrate that paternal tactile behaviors (i.e., SSC and ST) can have a favorable impact at biophysiological, behavioral, and psychological levels on fathers.

Similar to findings in a previous review [48,50], infants exposed to paternal SSC were able to maintain higher and more stable heart rates and a higher forehead temperature [38,40]. Furthermore, infants’ social and emotional behavior benefited from paternal SSC; infants’ crying lasted less time and they started breastfeeding behaviors earlier than control infants [40]. This is similar to previous results with mothers who provided SSC and ST to their infants [48,51]. Despite the limited number of studies and the focus on biophysiological variables, evidence about fathers’ and infants’ outcomes suggests that ST and SSC achieve similar outcomes.

A few studies have looked at differences and similarities between maternal and paternal ST by assessing parents’ biomarkers (i.e., oxytocin and testosterone) as outcome measures [31,33,36]. Feldman and colleagues [33] found that parent–infant tactile contact was associated with oxytocin level during parent-infant interaction, suggesting that oxytocin-mediated processes of paternal ST behaviors are comparable with those observed in mothers. However, differences between mothers and fathers emerged when the type of touch was taken into consideration. In mothers, oxytocin was associated with more affectionate touch during interactions, whereas oxytocin in fathers was related to increased positive arousal and stimulation [31]. Moreover, the interaction of oxytocin and testosterone predicted affectionate touch, so that high testosterone combined with high oxytocin was related to affectionate touch in mothers, but not in fathers [36]. These results suggest that the mechanism of action of testosterone is integrated with oxytocin in a gender-specific manner, namely, testosterone provides a biological background for oxytocin’s function as a modulator of maternal tactile behavior. Overall, these findings do not only highlight that specific ST (affectionate touch vs. stimulation touch) is characteristic of maternal and paternal caregiving behavior in the early stages of parenting, but also that the parent-specific mode of touch is related to a specific neuroendocrine profile.

Mixed findings emerged regarding the different impacts of paternal tactile behaviors on infants’ outcomes. Affectionate touch induces parasympathetic regulation in infants regardless of parental gender [39]. On the other hand, although no significant differences between paternal SSC and maternal SSC were found in the quiet state time, breastfeeding started later when SSC was provided by fathers [32,34,37]. These differences could be due to different methodological procedures, including settings (ecological context in SCC studies vs. experimental procedure in the RSA study, even if performed at home), type of parental tactile behavior (SSC vs. affectionate touch), and outcome variables measured (behavior, such as breastfeeding vs. parasympathetic regulation mediated by autonomic cardiorespiratory responses). Nevertheless, they could provide a picture of the impact of parental tactile behaviors on infants’ outcomes, including differences and similarities between mothers and fathers in the first months postpartum. On the one hand, maternal tactile behaviors measured by SSC were mainly related to breastfeeding, suggesting the primary role of body-to-body exchanges between mother and infant [52]. Parenthetically, this seems consistent with the gender-specific neurobiology underlying tactile behaviors and seems to be related to specific neuroendocrine profiles [31,33,36]. On the other hand, affectionate contact associated with regulation in infants appears to be a gender-neutral behavior useful for promoting the development of self-regulatory capacities in infants [39]. Thus, although fathers use more stimulation touch, the impacts on infants are comparable with those observed in mothers when they use affection touch. While these conclusions require further validation in larger samples, they have important cultural implications, as they provide evidence of the need to promote daily contact between fathers and infants during the first months of fatherhood.

### 4.1. Implications for Research

A number of studies have highlighted that fathers are now spending more time caring for, and interacting with, their children than in the past [53,54,55,56]. In the early months of their infant’s life, fathers do not only play a key role in supporting mothers [57,58] but they are also crucial for positive child outcomes [59,60,61,62]. In the latter context, paternal tactile behavior could provide an essential contribution. In spite of this, studies on this topic are still limited, therefore future research is needed to extend the study of paternal touch in typically developing children to beyond the first year of life. Although previous studies have examined the role of several biomarkers in parenting [9,63], more research on fathers is necessary. For example, given emerging research on the association between parental caregiving and DNA methylation [64], it would be of interest to examine the role of epigenetic mechanisms related to interpersonal tactile behavior in fathers. A recent review reported an association between maternal touch behavior and DNA methylation status [65], but to the best of our knowledge no study has yet considered it in fathers. Second, sociocultural norms and values may affect fathering [66] and, in turn, paternal tactile behaviors [67]. Of course, although sociocultural practices differ to some extent across societies, it still needs to be explored how different social views on fathering may be linked to specific patterns of paternal tactile behaviors and to the quality of touch. Third, the studies included in this review mainly focused on full-term typically developing children. While previous research examined the role of maternal touch in infants with neurodevelopmental disabilities [68], studies on clinical populations are needed to assess paternal tactile behavior in atypically developing children. This is closely related to understanding how paternal tactile behavior may be effective in promoting positive fathers’ caregiving and the socioemotional competence of children with neurodevelopmental disabilities. Thus, research efforts should be focused on the contribution of paternal tactile behaviors in parenting interventions. Likewise, the studies considered in this review do not specify the affective status of the father. Since previous research has stated that maternal touch is affected by the mother’s affective state [69,70], future studies might investigate whether the father’s affective state may have effects on touch behaviors directed toward the infant and what consequences might result, both for the father and the infant. Finally, while the papers reviewed have observed paternal touch behaviors in fathers and infants belonging to conventional families, touch behaviors in families which have a different setting (for example, LGBT, single-parent families…) have never been explored. Future research could implement some research questions on this topic.

### 4.2. Implications for Practice

This review demonstrates that paternal biophysiological, psychological, and behavioral outcomes related to tactile behaviors are similar to maternal ones. Consequently, a greater use of paternal tactile behaviors should be encouraged both in typical and atypical development in early childhood as well as in typical and atypical parenthood. Interventions promoting paternal social engagement may be recommended not only in typical contact conditions, so that the total amount of touch received by infants can be as much as possible, but also when mother–infant contact is reduced; for example, owing to postpartum depression. As affected mothers generally show minimal touch behaviors [70], interventions may aim to increase paternal touch behavioral repertoire towards infants in order to promote bonding [71]. Paternal touch could also be beneficial to families of full-term infants and at-risk infants (i.e., preterm), especially when mothers are unavailable due to clinical circumstances (e.g., maternal hospitalization).

### 4.3. Limitations

Some limitations of the review should be acknowledged, above all the paucity of studies involving fathers and limited studies comparing fathers’ and mothers’ tactile behaviors in full-term infants. Also, there is considerable variability among the included studies in terms of sample size, methodologies and the biomarkers used to assess outcomes. Furthermore, the available literature lacks systematic analysis of the possible role played by the infant sex in moderating the association between paternal touch and outcomes. Also, it should also be noted that studies were conducted in 10 different developed (Sweden, Italy, USA, Belgium, United Kingdom, Germany, and Israel) and developing countries (Chile, Taiwan, and Turkey). As sociocultural backgrounds may influence fathers’ approach to infants [66], the lack of studies examining the cultural aspects of fathering is yet another aspect that may limit the generalizability of findings to fathers of non-Western societies. In addition, the few available studies do not provide details about definitions and procedures, so that little is known about how long touch should be and how it should be provided by fathers. 

## 5. Conclusions

Overall, the findings of this review suggest that paternal tactile behaviors benefit both fathers and infants, and the evidence highlights that fathers’ touch is as pleasant as mothers’ touch for their infants. It is also important to note that mothers and fathers touch their infants in a different manner. While maternal touch appears to be characterized by affectionate touch, paternal touch is more related to increased positive arousal and stimulation [31,36]. From an evolutionary point of view, since humans are biparental species, this difference did possibly play a specific role in shaping the unique behavioral repertoires characteristic of maternal and paternal care. This, of course, does not imply that fathers do not use affectionate touch and mothers do not stimulate their infants through touch. 

Studying paternal tactile behavior holds the potential for improving both scientific knowledge and clinical applications. However, due to the many methodological challenges underlined in the current review, further research is warranted; for instance, to understand which aspects of touch may have an impact on infants’ development. Future studies should provide evidence about the role of fathers’ tactile behaviors, not only in typically developing children, but also in atypical development and in nonconventional families. New evidence may be useful for setting up specific early programs based on paternal tactile behaviors, and, ultimately, to inform parenting interventions for at-risk developmental conditions. More in general, the findings from the current review highlight the importance of stable physical contacts between fathers and their infants during early childhood in order to promote fathering.

## Figures and Tables

**Figure 1 behavsci-14-00060-f001:**
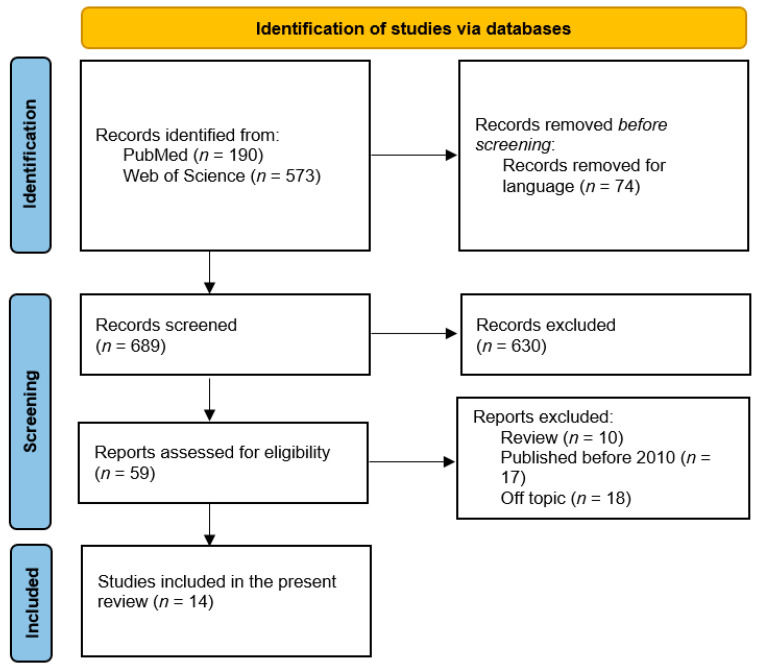
PRISMA flow diagram.

**Figure 2 behavsci-14-00060-f002:**
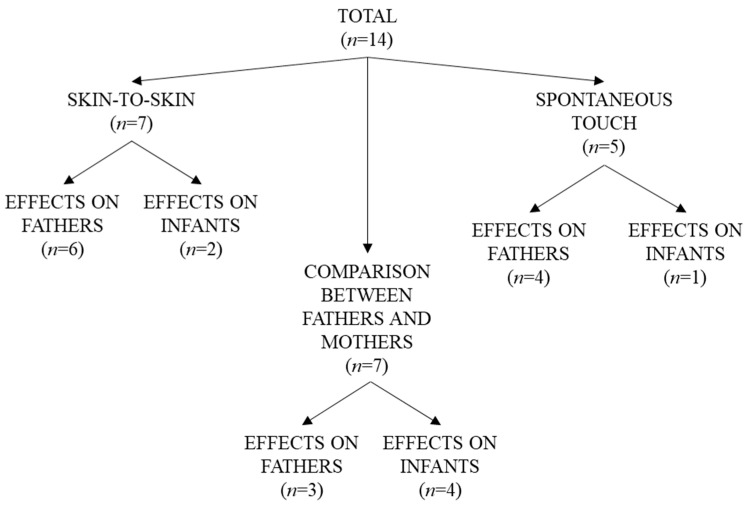
Classification and number of papers. Subtotals and overall totals of papers are lower than their sum within categories because papers were reported more than once when they met different categories.

**Table 1 behavsci-14-00060-t001:** Quality appraisal of the included studies. Labels: A—selection bias; B—study design; C—confounders; D—blinding; E—data collection methods; and F—withdrawals and drop-outs. Scoring: 3 = weak, 2 = moderate, 1 = strong, and NA = not applicable as the study does not have a suitable design. A final 1–3 score was assigned to each paper according to the presence of 2 or more weak scores (3—weak), only 1 weak score (2—moderate), or no weak scores (1—strong).

	Study	A	B	C	D	E	F	Final Score
1	2010, Feldman [31]	2	2	2	3	1	2	2
2	2010, Velandia [32]	2	1	2	3	1	1	2
3	2012, Feldman [33]	2	2	1	2	1	NA	1
4	2012, Velandia [34]	2	1	1	3	1	1	2
5	2014, Weisman [28]	2	2	1	2	1	3	2
6	2017, Chen [35]	2	1	1	2	1	NA	1
7	2017, Gordon [36]	2	2	2	3	1	NA	2
8	2017, Guala [37]	2	2	3	2	1	1	2
9	2019, Huang [38]	2	1	1	2	1	1	1
10	2019, Van Puyvelde [39]	2	2	1	3	1	1	3
11	2021, Ayala [40]	2	1	2	3	1	1	2
12	2021, Gettler [41]	2	2	1	3	1	3	3
13	2021, Morris [26]	2	2	1	3	1	1	2
14	2022, Yilmaz [42]	2	2	3	3	1	3	3

**Table 2 behavsci-14-00060-t002:** A summary of studies looking at fathers’ skin-to-skin care (SSC) and spontaneous touch (ST) with their infants.

First Author, Year	Country	Sample (*n*)	Infant Age (In Months)	Experimental Site	Type of Touch	Touch Coding	Variable Observed	Main Findings about Touch
Velandia, 2010 [32]	Sweden	72 infants (37 received SSC, 35 received standard care)	Newborns (38,75 gestational weeks, CI 95%)	Birthing room	SSC	None	Newborns and parents’ vocal interaction	Both fathers and mothers in SSC contact directed more soliciting sounds and speech to the infant and between them than did fathers and mothers without SSC contact. Infants who had SSC contact with their fathers cried significantly less than those in SSC contact with their mothers and shifted to a relaxed state earlier than in SSC contact with mothers.
Feldman, 2010 [31]	Israel	41 fathers	From 4 to 6 months old (166.3 ± 12.6 days)	Laboratory	ST	Microcoding including cradling, affectionate touch, proprioceptive touch and stimulatory touch	Physiological (salivary and plasma oxytocin)	Baseline levels of plasma and salivary oxytocin in mothers and fathers were similar, oxytocin levels in plasma and saliva were inter-related, and oxytocin was associated with the parent-specific mode of tactile contact. Human mothers who provided high levels of affectionate contact showed an oxytocin increase following mother–infant interaction but such increase was not observed among mothers displaying low levels of affectionate contact. Among fathers, only those exhibiting high levels of stimulatory contact showed an oxytocin increase.
Velandia, 2012 [34]	Sweden	37 infants	Newborn (just been born)	Birthing room	SSC	None	Strong rooting (i.e., breast-seeking behaviors, distinct head turning and movements, sometimes followed by smacking sounds), breast-massaging movements, breastfeeding, crying and the following parental behaviors and inter-active parental behaviors	Girls started rooting movements earlier than boys in SSC with either parent. Infants engaged in SSC with mothers started to breastfeed significantly earlier compared with SSC with fathers during the first 5–30 min. Girls cried more than boys in SSC with either parent. Mothers used more touching behavior towards their newborn infant than fathers. Mothers touched girls less than boys. Fathers directed less speech towards girls compared with boys.
Feldman, 2012 [33]	Israel	272 mothers and fathers and their infants, and 80 nonparents.	4-to-6-month-old	Laboratory	ST	Yes, but not specified.	Peripheral oxytocin, parental touch, gaze synchrony and parental care received in their own infancy.	Peripheral and genetic markers (i.e., oxytocin receptors and CD38 risk alleles) of the extended oxytocin pathway are interrelated and underpin core behaviors (i.e., parental touch and gaze synchrony) associated with human parenting and social engagement.
Weisman, 2014 [28]	Israel	35 fathers	5.01 ± 1.25 months old	Laboratory	ST during a wider interaction	Microcoding of parental touch (divided into affectionate touch; extremities—touching, which refers to touch the extremities of the infant’s body; or touch + object, which refers to touching the infant and playing with an object at the same time).	Physiological (testosterone levels mediated by administration of oxytocin) and observational (different aspects of interaction, including touch)	Lower baseline testosterone correlated with more positive father and infant behaviors. Oxytocin administration altered testosterone production in fathers, relative to the pattern of testosterone in the placebo condition. Finally, oxytocin-induced changes in testosterone levels correlated with parent–child social behaviors, including positive affect, social gaze, touch, and vocal synchrony.
Chen, 2017 [35]	Taiwan	92 fathers and their infants (46 received SSC and 46 received standard care)	Newborn (first three days of life)	Nursery	SSC	None	Father–child attachment measured by Father–Child attachment scale (FCAS)	The changes in the mean FCAS scores were found to be significantly higher in the group who eceived SSC than in the group who received standard care.
Guala, 2017 [37]	Italy	252 infants and their parents	Newborn	Birthing room	SSC	None	Duration of breastfeeding	A significant association between mother’s SSC and exclusive breastfeeding rates on discharge was found. This effect is maintained and statistically significant at three and six months, as compared to the groups that had paternal SSC care or no SSC care.
Gordon, 2017 [36]	Israel, USA, Germany	160 mothers and fathers (80 couples)	T0 = 1 months old (51.69 ± 14.65 days); T1= 6 months old (175.27 ± 31.65)	Home	ST		Physiological (plasma oxytocin and testosterone) and microcoding of interactions between each parent and infant	Paternal testosterone was individually stable across the first six months of parenting and predicted lower father–infant synchrony (i.e., parent engagement in social gaze, affectionate touch, and “motherese” vocalization while the infant looked at the parent and expressed positive affect). Testosterone has complex modulatory effects on the relations of oxytocin and parenting. Among fathers, only when testosterone was high, negative associations emerged between oxytocin and paternal affectionate touch.
Huang, 2019 [38]	China	108 fathers and their infants	Newborn	Hospital	SSC	None	Physiological (heart rate, forehead temperature), psychological (depression, anxiety and attachment) and behavioral (duration of crying, duration of breastfeeding)	Newborns in the treatment group had a more stable heart rate and forehead temperature, crying lasted less, and they started feeding behavior earlier. The duration of breastfeeding after SSC in the treatment group was longer as well. In addition, fathers in the treatment group had lower scores for anxiety and depression and better role attainment than those in the control group.
Van Puyvelde, 2019 [39]	Belgium and UK	50 infants	From 6 to 14 weeks old (10.40 weeks ± 2.63)	Home	ST	Stroking speed	Physiologic variables (heart rate, respiration rate, rr interval, respiratory sinus arrhythmia) and stroking rate	Infants’ respiratory sinus arrhythmia significantly increased during and after stroking, no matter whether touch was delivered by fathers or mothers. This effect was mediated by both heart rate and respiration. However, respiratory mediation occurred later when delivered by fathers than by mothers.
Gettler, 2021 [41]	USA	211 fathers	T0 = newborn; T1 = 2 to 4 months old (11.6 ± 6.54 weeks)	Birthing unit and home	SSC	None	Physiological (oxytocin)	First-time fathers’ oxytocin was higher following first holding of their newborns, compared to their preholding levels. Contrasting with prior results, fathers’ post-holding oxytocin levels following SSC did not differ from preholding levels, whereas fathers who provided standard holding showed higher oxytocin post-holding.
Ayala, 2021 [40]	Sweden, Chile	95 infants (32 received standard care in the cot, 34 experienced touch in their fathers’ arms and and 29 received SSC)	Newborn (gestational age 38.9 ± 0.9 weeks)	Neonatal unit	SSC	None	Physiological (body temperature, heart rate and oxygen saturation) and their wakefulness (by using the Neonatal Behavioral Assessment Scale)	Heart rates were significantly higher in the SSC than cot or fathers’ arms groups and showed greater stability over time. Wakefulness was initially higher in the SSC group, but there were no significant differences by the end of the observation. There were no differences between the groups in peripheral oxygen saturation. SSC contact had no negative impact on infants.
Morris, 2021 [26]	USA	45 fathers	6 months old (6.61 ± 0.46 months)	Laboratory	ST	Microcoding of paternal physical touch at 1/10 s intervals during a laboratory-based free-play interaction	Physiological (oxytocin)	Fathers who engaged in more playful proprioceptive touch showed higher levels of oxytocin. Gentle affectionate touch and functional proprioceptive touch assiociated with higher unextracted oxytocin levels. Fathers who did not show physical touch had lower levels of both unextracted and extracted oxytocin.
Yilmaz, 2022 [42]	Turkey	69 fathers (34 who established SSC with their infants and 35 who did not come into SSC with their infants).	T0 = up to three hours after birth T1 = 6 to 12 months old	Maternity ward and home	SSC	None	Father–infant attachment measured by Paternal-Infant Attachment Scale	The total score of fathers who established skin-to-skin contact with their babies was significantly higher than that of the control group. Moreover, the questionnaire filled in by first-time fathers was higher than that of the control group.

## Data Availability

No new data were created or analyzed in this study. Data sharing is not applicable to this article.

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
