# Peer review of "Skin-to-Skin Care and Spontaneous Touch by Fathers in Full-Term Infants: A Systematic Review"

_behavsci, 2024, doi:10.3390/bs14010060_

Round 1

Reviewer 1 Report

Comments and Suggestions for Authors

Thank you for the opportunity to review this fascinating manuscript. The authors are clear that a paucity of research exists on fathers' experience with SSC and the full-term newborn. However, a few of the articles cited focused on infants from 1-12 months which does not support the aims of this study. If one excluded these toddler variables, the "N" would be even smaller. The article needs to stay focused on "SSC and the full-term infant." Generalizability would not be significant even with including these other studies. My recommendation is for the authors to remove the other citations, focus on what evidence there is, and keep their passion of getting more research in this area as a "Call to Action."

Comments on the Quality of English Language

The English was not a problem in this article.

Author Response

Reviewer #1

Dear Reviewer,

We respond to the comments in their original order, explaining on a point-by-point basis how we have addressed each point. Please note that in the revised version of the manuscript all changes are highlighted.

Thank you for your consideration.

Comment #1: Thank you for the opportunity to review this fascinating manuscript.

Reply#1: We thank the reviewer for the positive comment. Also, thank you very much for your time spent in reviewing our manuscript, for your consideration and for highlighting the relevance of our study.

Comment #2: The authors are clear that a paucity of research exists on fathers' experience with SSC and the full-term newborn. However, a few of the articles cited focused on infants from 1-12 months which does not support the aims of this study. If one excluded these toddler variables, the "N" would be even smaller. The article needs to stay focused on "SSC and the full-term infant." Generalizability would not be significant even with including these other studies. My recommendation is for the authors to remove the other citations, focus on what evidence there is, and keep their passion of getting more research in this area as a "Call to Action."

Reply#2: The reviewer raised a relevant point. For what concerns Ünal Toprak et al., 2021, we agree with the reviewer. Indeed, not only the final measure was conducted when the infants were 12 months-old but also paternal skin-to-skin contact was carried out not only in the perinatal period but during the entire first year of infant life and this amount of touch differ significantly from what happened in the other studies included in the review. Thus, as the reviewer suggested we have now eliminated this paper from the final pool of the study and fixed the manuscript accordingly. Differently, we believe that Yilmaz et al., 2022, need to be included in the final pool of the study of this review. Please, let us explain:  although the touch effects were followed-up until infants were 12 months old, the touching behavior occurred in the perinatal period (first hours of life).  As a consequence, it is appropriated including it in our revision, given that this work investigated the impact of early paternal touch on infant’s mid-term outcomes. We have now better specify this aspect in the paper as follow: “Yilmaz and colleagues [38] showed that similar findings are true also in a middle-term perspective. They compared father-infant attachment – as measured by the Paternal-Infant Attachment Scale [39] – in fathers during SSC immediately after birth and fathers who did not engage in SSC after 6-to-12 months after birth. Attachment was more intense in fathers who engaged in SSC, especially in first-time fathers”.

Reviewer 2 Report

Comments and Suggestions for Authors

I sincerely appreciate the authors' efforts to bring light to the effects of skin-to-skin care (SSC) and spontaneous touch by fathers in full-term infants in the literature. The authors conducted a systematic review of the biological, behavioral, and psychological impact of paternal SSC and spontaneous touch and then examined the differences in the effects of fathers' and mothers’ tactile caregiving behavior. I believe that this manuscript can provide meaningful guidance for researchers, but several points of clarification should be addressed to maximize the impact of this manuscript. Specifically, the main concerns lie in the author's synthetization of the literature.

Introduction

1.     I appreciate the author’s conceptualization of the term SSC. I wonder if this practice is more common among certain cultures, or it is a universal parenting practice. Please provide citations and clarification to give readers more context.

2.     The definition of spontaneous touch should come earlier in the first time you introduce this concept (line 53).

Materials and Methods

1.     Why did the author set the timeline between 2010 to 2023? More clarification is needed.

2.     In the 2.4 section (line 106) – screening for language. What does this mean? Does it mean screening for language other than English, or language about the SCC/ spontaneous touch literature?

Results

1.     I suggest combining 3.1 and 3.2 into one section as here are many overlaps between them.

2.     I recommend that the author provide some more context about the study characteristics in the 3.2 paragraph, (e.g., the country in which the studies were conducted, race/ethnicity of the families, and sex of infants/toddlers).

3.     Table 2 is now spelled as Tabella 2. Please correct.

4.     In Table 2, there are multiple studies that utilized a randomized control trial nature – please elaborate on the “experimental group” vs “control group” or “placebo group”. It is unclear what interventions were used and what is considered as control.

5.     For the main findings in Velandia (2010), what does “more vocally” mean? More verbal exchange between parents? More verbal description of their child? Please be more specific.

6.     For the main findings in Valandia (2012), please provide a simple definition of “rooting”.

7.     For the main findings in Feldman (2012), “Peripheral and genetic markers of the extended oxytocin pathway are interrelated and underpin core behaviors associated with human parenting and social engagement” What genetic markers were included, please specify. Also, “underpin core behaviors” – what core behaviors are we referring to? Please be more specific on the findings.

8.     Similarly, for the main findings of Weisman (2014), the authors mentioned “more optimal father and infant behaviors” but it is unclear what is considered optimal. Testosterone should be spelled out and not referred to as “T” in findings. In the coding, authors noted” extremities – touching infant’s extreme part of the body”, this is confusing as I don’t understand what this means.

9.     For Guala's (2017) main findings, I recommend rewording the first sentence to “A significant association between mother’s SSC and breastfeeding rates.” However, I had a hard time understanding what “exclusive breastfeeding means” and whether is it “on discharge or post-discharge”. Please clarify.

10.  For Gordon’s (2017) main findings, what does “father-infant synchrony” mean? The authors mentioned about micro-coding of “father-infant interaction” but it was not described what was considered synchrony” (e.g., more touch between father and infant).

11.  Please elaborate on the moderation findings of oxytocin and testosterone on fathers’ post-partum involvement and bonding in Gettler’s (2021) findings.

12.  Ayala (2021) main findings, “cot” do you mean “control”? Please check spelling and inappropriate abbreviations throughout Table 2.

13. Yilmaz's (2022) main findings, the authors alluded that there are two groups given he mentioned a control group but did not elaborate on the description of participants as now it only reads “69 fathers”.

14.  Please also reflect on all changes I recommended for Table 2 to the results text, more context is needed to fully understand the literature and have a more accurate synthesis and interpretation.

15.  (line 159) Authors mentioned depression and anxiety in one study, which needed citation and elaborations of findings. However, I did not see any mention of depression and anxiety in the main findings of Table 2, please add that in.

Discussion

1.     Please reword this sentence “Like mothers involved in SSC [49] and similarly to fathers who provided spontaneous touch, specifically affectionate touch [25], fathers benefitted in terms of attaining paternal role and better interaction 299 with their infants, and were less stressed and anxious [32,33,38,39].” (line 297-299) It is long and hard to follow, making it confusing for readers.

2.     In lines 300-301, authors noted that “bio-physiological markers such as oxytocin concentration changed after interaction with their infants [25,42], although this effect could be mediated by testosterone levels [31,43].” How did the concentration change – increased/decreased? What does this change imply? The mediation pattern should be elaborated and explained with implications of this.

3.     I recommend authors to reorganize the flow of the discussion section (lines 290 – 354)– start by summarizing bio-physiological findings (e.g., oxytocin, cardiac activity), then follow by psychological and behavioral. Please elaborate on the implications of these findings – what do these results mean?

4.     In Section 4.2, the authors mentioned “Interventions promoting paternal social engagement may be recommended when maternal-infant contact is reduced, for example owing to postpartum depression.” I think that authors should also recommend paternal engagement with infants even if material-infant contact is established, not only when maternal-infant contact is reduced.

5.     The limitation section could be more robust. Are the studies only generalizable to certain family structures (what about single parenthood, LGBTQ parents)? Would it be important to establish whether SCC during newborn is more impactful than other developmental time points? Does the positive impact of touch extend to other development periods, such as early childhood and middle childhood? Does infant sex play a role in moderating the association between parental touch and outcomes?

Comments on the Quality of English Language

adequate

Author Response

Dear Reviewer,

We respond to the comments in their original order, explaining on a point-by-point basis how we have addressed each point. Please note that in the revised version of the manuscript all changes are highlighted.

Thank you for your consideration.

Reviewer #2

Comment #1: I sincerely appreciate the authors' efforts to bring light to the effects of skin-to-skin care (SSC) and spontaneous touch by fathers in full-term infants in the literature. The authors conducted a systematic review of the biological, behavioral, and psychological impact of paternal SSC and spontaneous touch and then examined the differences in the effects of fathers' and mothers’ tactile caregiving behavior. I believe that this manuscript can provide meaningful guidance for researchers, but several points of clarification should be addressed to maximize the impact of this manuscript. Specifically, the main concerns lie in the author's synthetization of the literature.

               Reply#1: We thank the reviewer for the positive comment on our work. We replied to all reviewer concerns point-by-point to improve our manuscript following the reviewer suggestions.

Comment #2: Introduction - I appreciate the author’s conceptualization of the term SSC. I wonder if this practice is more common among certain cultures, or it is a universal parenting practice. Please provide citations and clarification to give readers more context.

Reply#2: As suggested by clinical practice as well as several studies SSC is a cross-cultural practice (Abdulghani et al., 2018).  It is a very low-cost practice that does not require specific materials and equipment, and  it is suitable for all cultural backgrounds. The practice was initially conducted in Bogotà (Colombia) by Edgar Rey and Hector Martinez, in the late 1970s. It is a very well suited strategy to address neonatal and infant mortality and morbidity in low technical resources settings: SSC (KMC) is an effective way to meet the baby’s needs for warmth, breastfeeding, protection from infection, stimulation and safety. The SSC has been approved and supported by the World Health Organization and is currently considered the "gold standard" in the care for premature and/or low birth weight. In low-resource countries, SSC has been shown to contribute to significantly reducing the high mortality and morbidity rate in children born prematurely. Following the reviewer comment we have better specify this aspect as follow: “SSC is a  practice where a naked infant is placed on his/her parent’s bare chest. It is commonly used immediately after birth with full-terms, or any time an infant needs be comforted or calmed down. If necessary, infants can wear a diaper and/or a cap and par-ents and infants can be covered with a blanket or linen. Importantly, given that  Since it is a very low-cost practice that does not require specific materials and equipment, it is suitable for all cultural backgrounds (Abdulghani et al., 2028)”

Comment #3: Introduction - The definition of spontaneous touch should come earlier in the first time you introduce this concept (line 53).

Reply#3: We thank the reviewer for the suggestion. We have now moved the definition of spontaneous touch as suggested.

Comment #4: Materials and Methods - Why did the author set the timeline between 2010 to 2023? More clarification is needed.

Reply#4: We know this is an arbitrary choice, but we believe that this period of time ensures the review to be as up-to-date as possible. Furthermore, some of the previous papers are very dated (for example, Rödholm & Larsson, 1979), and fathers have only recently been involved in perinatal care systematically (Kaźmierczak & Karasiewicz, 2019). . We now specified this point as follow: “This time period was chosen to ensure the novelty of the review and because fathers have only recently been involved in perinatal care (Kaźmierczak et al., 2019)”.

Comment #5: Materials and Methods - In the 2.4 section (line 106) – screening for language. What does this mean? Does it mean screening for language other than English, or language about the SCC/ spontaneous touch literature?

Reply#5: We are sorry that it was not clear to the reviewer. It means screening for language other than English, we better specified it in the manuscript.

Comment #6: Results - I suggest combining 3.1 and 3.2 into one section as here are many overlaps between them.

Reply#6: We have now combine 3.1 and 3.2 as suggested by the reviewer.

Comment #7: Results - I recommend that the author provide some more context about the study characteristics in the 3.2 paragraph, (e.g., the country in which the studies were conducted, race/ethnicity of the families, and sex of infants/toddlers).

Reply#7: We have now improved the mentioned paragraph (3.1 in the revised manuscript) as suggested by the reviewer. In particular, we modify the paragraph as follow: “Children's age spans from a minimum of 1 months (i.e., newborns) to a maximum of 12 months. All of the studies included both males and females and were conducted in different cultural background.”

Comment #8: Results - Table 2 is now spelled as Tabella 2. Please correct.

Reply#8: Fixed.

Comment #9: results - In Table 2, there are multiple studies that utilized a randomized control trial nature – please elaborate on the “experimental group” vs “control group” or “placebo group”. It is unclear what interventions were used and what is considered as control.

Reply#9: We better specify this according to reviewer comment.

Comment #10: Results - For the main findings in Velandia (2010), what does “more vocally” mean? More verbal exchange between parents? More verbal description of their child? Please be more specific.

Reply#10: We have now better clarify what it means as follow: “Both fathers and mothers in SSC contact directed more soliciting sounds and speech to the infant and between them than did fathers and mothers without SSC contact”

Comment #11: Results - For the main findings in Valandia (2012), please provide a simple definition of “rooting”.

Reply#11: We have now entered a definition as follow “Strong rooting ( i.e. breast-seeking behaviours, distinct head turning and movements, sometimes followed by smacking sounds)”

Comment #12: Results - For the main findings in Feldman (2012), “Peripheral and genetic markers of the extended oxytocin pathway are interrelated and underpin core behaviors associated with human parenting and social engagement” What genetic markers were included, please specify. Also, “underpin core behaviors” – what core behaviors are we referring to? Please be more specific on the findings.

Reply#12: Now we have better specified what the reviewer asked, both in the text and in the table, as follow “Peripheral and genetic markers (i.e., oxytocin receptors and CD38 risk alleles) of the extended oxytocin pathway are interrelated and underpin core behaviours (i.e., parental touch and gaze synchrony) associated with human parenting and social engagement“.

Comment #13: Results - Similarly, for the main findings of Weisman (2014), the authors mentioned “more optimal father and infant behaviors” but it is unclear what is considered optimal. Testosterone should be spelled out and not referred to as “T” in findings. In the coding, authors noted” extremities – touching infant’s extreme part of the body”, this is confusing as I don’t understand what this means.

Reply#13: We have better specified and rephrased the sentences according to the reviewer’s suggestions.

Comment #14: Results - For Guala's (2017) main findings, I recommend rewording the first sentence to “A significant association between mother’s SSC and breastfeeding rates.” However, I had a hard time understanding what“exclusive breastfeeding means” and whether is it “on discharge or post-discharge”. Please clarify.

Reply#14: Following the reviewer’s suggestion as for the Guala's (2017) main findings, we have rephrased the sentence.

As suggested by WHO (https://www.who.int/tools/elena/interventions/exclusive-breastfeeding) exclusive breastfeeding means that the infant receives only breast milk; no other liquids or solids are given – not even water – with the exception of oral rehydration solution, or drops/syrups of vitamins, minerals or medicines. We have addressed this by adding a short clarification. 

As regard “on discharge or post-discharge”, it means that the results appear significant both on the exact time of discharge and at three and six months after. As the sentence sounds clear enough we respectfully disagree so that we consider that the actual wording works.    

Comment #15: Results - For Gordon’s (2017) main findings, what does “father-infant synchrony” mean? The authors mentioned about micro-coding of “father-infant interaction” but it was not described what was considered synchrony” (e.g., more touch between father and infant).

Reply#15: As reported in the revised paragraph 3.3.1, synchrony is meant as “parent engagement in social gaze, affectionate touch, and “baby-talk” vocalization while the infant looked at the parent and expressed positive affect”. We added this specification in the Table 2 as well.

Comment #16: Results - Please elaborate on the moderation findings of oxytocin and testosterone on fathers’ post-partum involvement and bonding in Gettler’s (2021) findings.

Reply#16: We have specified it in Table XX as follow: “Fathers whose testosterone increased while first holding their newborns and who experienced greater concomitant decreases in oxytocin reported more involvement in direct caregiving and greater father-infant bonding months later, compared to men who testosterone and oxytocin both declined during holding”.

Comment #17: Results - Ayala (2021) main findings, “cot” do you mean “control”? Please check spelling and inappropriate abbreviations throughout Table 2.

Reply#17: ”Cot” is the term used by Ayala and colleagues, by which they mean the small bed for infants or very young child.  As the word seems clear enough, we respectfully disagree with the request to further specify this term.

Comment #18: Results - Yilmaz's (2022) main findings, the authors alluded that there are two groups given he mentioned a control group but did not elaborate on the description of participants as now it only reads “69 fathers”.

Reply#18: The sample consisted of 69 fathers (34 who established SSC with their infants and 35 who did not come into SSC with their infants). Now we have added this information in Table 2.

Comment #19: Results - Please also reflect on all changes I recommended for Table 2 to the results text, more context is needed to fully understand the literature and have a more accurate synthesis and interpretation.

Reply#19. In the text, we implemented with the reviewer's suggestions the points where the results seemed less clear and we could add clarifications.

Comment #20: Results - (line 159) Authors mentioned depression and anxiety in one study, which needed citation and elaborations of findings. However, I did not see any mention of depression and anxiety in the main findings of Table 2, please add that in.

Reply#20: As reported in the text, the elaboration of Huang and colleagues is “Similarly, one study recruited full-term healthy infants born by Caesarean section and fathers divided into two groups: SSC vs. routine care [35]. Fathers filled out self-report questionnaires about anxiety, depression and role attainment. Fathers who engaged in SSC had lower anxiety/depression and higher paternal role attainment scores after SSC”. You can find it also in Table 2 as follow “In addition, fathers in the treatment group had lower scores of anxiety and depression and better role attainment than those in the control group”. We added a specification in the column “variable observed”.

Comment #21: Results - Please reword this sentence “Like mothers involved in SSC [49] and similarly to fathers who provided spontaneous touch, specifically affectionate touch [25], fathers benefitted in terms of attaining paternal role and better interaction 299 with their infants, and were less stressed and anxious [32,33,38,39].” (line 297-299) It is long and hard to follow, making it confusing for readers.

Reply#21: Following the Reviewer’s suggestion, we have rephrased the sentence in order to make it simpler as follow “Lastly, like mothers involved in SSC [48] and similarly to fathers who provided affectionate ST [26], fathers benefitted in terms of attaining paternal role and better interaction with their infants, and were less stressed and anxious [33–35,49].

Comment #22: Results - In lines 300-301, authors noted that “bio-physiological markers such as oxytocin concentration changed after interaction with their infants [25,42], although this effect could be mediated by testosterone levels [31,43].” How did the concentration change – increased/decreased? What does this change imply? The mediation pattern should be elaborated and explained with implications of this.

Reply#22: We have made the following edits to better illuminate the findings: Despite this, it emerged that paternal touch can have positive effects on both fathers and infants, both with regard to SSC and ST. Indeed, there is a relation between paternal bio-physiological markers such as oxytocin and father-infant tactile interaction [26,31,32,43], although this relation can change in light of the experimental design. Some evidence suggests that higher basal oxytocin predicts a higher number of paternal tactile behaviour [31], other that tactile behaviours increase the level of post-interaction oxytocin [26]. This association could be influenced by other significant paternal variables (i.e., parity status and testosterone levels). In particular, in Gettler et al. [32] it was reported that first-time fathers' oxytocin was higher following first holding of their newborns, compared to their pre-holding levels. Gordon et al. [43] reported that the association between oxytocin and father-infant tactile interaction (i.e., affectionate touch) was moderated by another bio-physiological marker (i.e., testosterone): only in the fathers with high level of testosterone, oxytocin was negatively associated with paternal affectionate touch .

Comment #23: Results - I recommend authors to reorganize the flow of the discussion section (lines 290 – 354)– start by summarizing bio-physiological findings (e.g., oxytocin, cardiac activity), then follow by psychological and behavioral. Please elaborate on the implications of these findings – what do these results mean?

Reply#23: We reorganized the discussion as suggested by the reviewer, keeping symmetry with the paragraph of Results (i.e.: bio-physiological, behavioural and psychological). Implications for research and practice are discussed in specific sections, respectively 4.1 and 4.2.

Comment #24: Results - In Section 4.2, the authors mentioned “Interventions promoting paternal social engagement may be recommended when maternal-infant contact is reduced, for example owing to postpartum depression.” I think that authors should also recommend paternal engagement with infants even if material-infant contact is established, not only when maternal-infant contact is reduced.

Reply#24: The reviewer raises an important point. We are grateful to reviewer for encouraging us to add this aspect. We have addressed this by adding the following sentence: “Consequently, a greater use of paternal tactile behaviours should be encouraged both in typical and atypical development in early childhood and in typical and atypical parenthood. Interventions promoting paternal social engagement may be recommended not only in typically developing children, so that the total amount of touch received by infants can be as much as possible, but also when mother-infant contact is reduced, for example owing to postpartum depression”.

Comment #25: The limitation section could be more robust. Are the studies only generalizable to certain family structures (what about single parenthood, LGBTQ parents)? Would it be important to establish whether SCC during newborn is more impactful than other developmental time points? Does the positive impact of touch extend to other development periods, such as early childhood and middle childhood? Does infant sex play a role in moderating the association between parental touch and outcomes?

Reply#25: The reviewer makes the following main points:

(1) Are the studies only generalizable to certain family structures (what about single parenthood, LGBTQ parents)?

(2) Would it be important to establish whether SCC during newborn is more impactful than other developmental time points?

(3) Does the positive impact of touch extend to other development periods, such as early childhood and middle childhood?

and

(3) Does infant sex play a role in moderating the association between parental touch and outcomes?

Regarding (1):  To the best of our knowledge there is not data on the association between family structures mentioned by reviewer (e.g., single parenthood, LGBTQ parents) and touch. That being said, usually limitation section should mention and discuss the weaknesses as well as the strengths of the studies revised, including for example issues with research samples and selection, insufficient sample size for statistical measurements, methods/instruments/techniques used to collect the data. Lack of previous research studies on related topics is an important point, but the reviewer’s proposal could be more mentioned as a part of potential future research. In order to better illuminate this point (and other ones, see below) we have added a sentence in the Implications for research section (“Finally, revised paper have observed paternal touch behaviours in fathers and infants belonging to a conventional families, touch behaviours in families which have a different setting (for example, LGBT, single-parent families…) have never been explored. Future research could implement some re-search questions about this topic”.

Regarding (2 & 3): As the points 2 and 3 are very related topics we discuss them together. Research on maternal touch suggest that the mother’s touch evolves with the infant’s age; maternal affectionate and stimulating touch, for example, seem to decrease significantly during the second 6 months of life (Ferber et al., 2008). However, it should be noted that studies of maternal touch in typically developing infants have mainly focused on the first months of life which seems to be critical for the individual development (Pelaez-Nogueras et al., 1996). On the other hand, beneficial effects of touch on physiological regulation and behavioral outcomes have been identified throughout the lifespan. At birth, affective touch stimulation reduces physical arousal and heart rate in healthy infants. Even in an at-risk condition such as prematurity, gentle-touch significantly improves physiological regulation, and promotes early affiliative behaviors and parent-infant bonding. During the transition to toddlerhood and preschool, the critical role of touch expands beyond caregivers and immediate family to include teachers and peers. As children get older and more independent, the sphere narrows again and children receive tactile input from fewer people and in fewer contexts than when they were very young (Cascio, Moore and McGlone, 2019). During childhood, the frequency of touch enacted with parents and siblings has been associated with the strength of connectivity of the posterior superior temporal sulcus and other nodes of the social brain and was predictive of children’s expression of positive emotions. In adolescence and adulthood, affective touch appears in new facets such as those related to romantic and sexual attraction. In sum, individual exposure to affective touch is associated with different aspects of well-being throughout the life span. However, as mentioned, the effects of touch in early phase of life in which infants require more closeness with their caregiver. That being said, considering the role of paternal touch again the issues raised by the reviewer seem potential questions research. As a consequence, we have addressed this by adding some sentences in Implications for research section. Specifically, we have focused on the future research about the impact of positive paternal touch in other development periods and added a sentence about this topic.

“In spite of this, studies on this topic are still limited, therefore future research is needed, to extend the study of paternal touch in typical developmental conditions and beyond the first year of life.”

Regarding (4): Infants’ sex has been shown to influence caregiving behaviors, including touch; for example, mothers of sons seem to touch their infants more often and use more stimulating or moving touch than mothers of daughters (Fausto-Sterling et al., 2015). Actually, we have mentioned this aspect if it was discussed by authors. Indeed, some of reviewed  papers (e.g.: Velandia et al., 2012; Feldman et al., 2012) report that the infants’s sex plays a role in moderating parental touch behaviours. These results are reported in Table 2 and in the Results section. However, the lack of systematic analyses in the reviewed studies about the role played by the infant sex in moderating the association between paternal touch and outcomes sounds like an important limitation. Now we have added a sentence to illuminate this limitation: “Furthermore, the available literature lacks of a systematic analysis about the possible role played by the sex of the infant in moderating the association between paternal touch and outcomes.

Reviewer 3 Report

Comments and Suggestions for Authors

The study deals with a relevant and highly topical subject in the field of research. There is still a gap in studies on fathers' emotional relationships with their young children, and as this research has shown, few studies deal with touch between fathers and their infants. It is very interesting to see that the interaction pattern between fathers and their children is changing in terms of both quality and quantity (time spent with the child), as mentioned in the article. It's even more important to see that touch (both spontaneous touch and skin-to-skin touch) has positive consequences for both the father and the infant, as has been studied at the psychological, behavioural and endocrine levels. Even though in the case of mother-baby touch there is still some advantage in improving breastfeeding conditions, the authors were still able to demonstrate positive effects on biomarkers and psychological variables as well as on the behavioural level. It would be interesting to carry out a future study to assess cultural differences in parent-baby relationships, as mentioned in the "research implications" section, and perhaps it would be important to question or deepen the quality of touch. This is a well-described systematic review, with a transparent and judicious search procedure. It was not clear to me why studies before 2010 were not included. However, this does not affect the quality of the work. 

The objectives are well described: (1) to carry out a systematic review on the impact of paternal touch behaviour (spontaneous and skin-to-skin) on both the infant and the father from a behavioural, physiological and psychological perspective. And the second objective was to examine the evidence from studies that compared the differences in touch on babies between fathers and mothers 

The authors managed to answer the initial objectives and organised the discussion into topics covering the results for skin-to-skin and spontaneous touch and separating out the impacts for fathers and their babies. They also separated the results into sub-items discussing physiological, psychological and behavioural impacts. The authors found very interesting results and systematised them to show that there was a positive impact of touch (both skin-to-skin and spontaneous) for both parents and babies. Parents who did skin-to-skin showed higher levels of oxytocin, greater vocalisation towards the baby and more attachment, therefore with results at all three levels of evaluation. For the babies, the results were in the same direction - they benefited more in terms of heart rate stability, cried less and started breastfeeding earlier. With regard to spontaneous touch, the authors found less research and could only answer the physiological impact - with higher levels of oxytocin for the parents and higher RSA in the babies. They found no research that discussed the behavioural and psychological impact in the case of studies on spontaneous touch. With regard to objective two, the authors showed that babies benefit from both maternal and paternal touch, but there are also endocrine differences between maternal and paternal reactions. 

The discussion is well directed, describing all the studies in an organised way. The authors also complete the study by discussing the implications of these results.

I have just one small suggestion:  

1. line 151 "Two studies observed behavioural variables during skin-to-skin contact, understanding them as vocal interaction: I suggest "Two studies analysed vocalisation behaviour during SCC contact"

Comments on the Quality of English Language

I had no difficulty understanding the text written in English and I didn't find any mistakes, even though my first language is not English.

Author Response

Dear Reviewer,

We respond to the comments in their original order, explaining on a point-by-point basis how we have addressed each point. Please note that in the revised version of the manuscript all changes are highlighted.                   

Thank you for your consideration.

Comment #1: The study deals with a relevant and highly topical subject in the field of research. There is still a gap in studies on fathers' emotional relationships with their young children, and as this research has shown, few studies deal with touch between fathers and their infants. It is very interesting to see that the interaction pattern between fathers and their children is changing in terms of both quality and quantity (time spent with the child), as mentioned in the article. It's even more important to see that touch (both spontaneous touch and skin-to-skin touch) has positive consequences for both the father and the infant, as has been studied at the psychological, behavioural and endocrine levels. Even though in the case of mother-baby touch there is still some advantage in improving breastfeeding conditions, the authors were still able to demonstrate positive effects on biomarkers and psychological variables as well as on the behavioural level.

Reply#1: We thank the reviewer for the positive comments. Also, thank you very much for your time spent in reviewing our manuscript, for your detailed consideration and for highlighting the relevance of our study.

Comment #2. It would be interesting to carry out a future study to assess cultural differences in parent-baby relationships, as mentioned in the "research implications" section, and perhaps it would be important to question or deepen the quality of touch.

Reply#2: Thanks to the reviewer for suggesting us to add the aspect about the quality of touch in different cultures. We added some words about this topic: “Of course, although socio-cultural practices differ to some extent across societies, it still needs to be explored how different social views on fathering may be linked to specific patterns of paternal tactile behaviours and to the quality of touch.

Comment #3: This is a well-described systematic review, with a transparent and judicious search procedure. It was not clear to me why studies before 2010 were not included. However, this does not affect the quality of the work. 

Reply#3:We know this is an arbitrary choice, but we believe that this period of time ensures the review to be as up-to-date as possible Furthermore, some of the previous papers are very dated (for example, Rödholm & Larsson, 1979), and fathers have only recently been involved in perinatal care systematically (Kaźmierczak & Karasiewicz, 2019). This time period was chosen to ensure the novelty of the review and because fathers have only recently  been involved in perinatal care . We now specified this point as follow: “This time period was chosen to ensure the novelty of the review and because fathers have only recently been involved in perinatal care (Kaźmierczak et al., 2019)

Comment #4. The objectives are well described: (1) to carry out a systematic review on the impact of paternal touch behaviour (spontaneous and skin-to-skin) on both the infant and the father from a behavioural, physiological and psychological perspective. And the second objective was to examine the evidence from studies that compared the differences in touch on babies between fathers and mothers. The authors managed to answer the initial objectives and organised the discussion into topics covering the results for skin-to-skin and spontaneous touch and separating out the impacts for fathers and their babies. They also separated the results into sub-items discussing physiological, psychological and behavioural impacts. The authors found very interesting results and systematised them to show that there was a positive impact of touch (both skin-to-skin and spontaneous) for both parents and babies. Parents who did skin-to-skin showed higher levels of oxytocin, greater vocalisation towards the baby and more attachment, therefore with results at all three levels of evaluation. For the babies, the results were in the same direction - they benefited more in terms of heart rate stability, cried less and started breastfeeding earlier. With regard to spontaneous touch, the authors found less research and could only answer the physiological impact - with higher levels of oxytocin for the parents and higher RSA in the babies. They found no research that discussed the behavioural and psychological impact in the case of studies on spontaneous touch. With regard to objective two, the authors showed that babies benefit from both maternal and paternal touch, but there are also endocrine differences between maternal and paternal reactions. The discussion is well directed, describing all the studies in an organised way. The authors also complete the study by discussing the implications of these results.

Reply#4: Thank you for your careful and accurate reading. Also thank you for all your positive comment on our work.

Comment #5: I have just one small suggestion: 1. line 151 "Two studies observed behavioural variables during skin-to-skin contact, understanding them as vocal interaction: I suggest "Two studies analysed vocalisation behaviour during SCC contact"

Reply#5: Thank you for your suggestion, we rephrased the sentence according to your advice.

Round 2

Reviewer 2 Report

Comments and Suggestions for Authors

I appreciate the authors' effort in making edits based on the comments. I think that the paper is significantly improved. I appreciate the authors effort in contributing to the field. Nevertheless, I am still detecting typos throughout the paper (for example Table 2 Velandia 2010 under-sample - "receuved" care). I urge the authors to re-read and correct typos throughout the paper. 

Comments on the Quality of English Language

Good.

Author Response

Dear Reviewer,

We greatly appreciate your attention in reading the revised version of our manuscript.                   

We respond to the comments in their original order, explaining on a point-by-point basis how we have addressed each point.                    

Thank you for your consideration. 

Reviewer 2

Comment #1: I appreciate the authors' effort in making edits based on the comments. I think that the paper is significantly improved. I appreciate the authors effort in contributing to the field.

Reply #1: We thank the reviewer for the positive comment.

Comment #2: Nevertheless, I am still detecting typos throughout the paper (for example Table 2 Velandia 2010 under-sample - "receuved" care). I urge the authors to re-read and correct typos throughout the paper.

Reply #2: We re-read the manuscript and corrected the detected typos.
